# Beverage Intake and Drinking Patterns—Clues to Support Older People Living in Long-Term Care to Drink Well: DRIE and FISE Studies

**DOI:** 10.3390/nu11020447

**Published:** 2019-02-21

**Authors:** Oluseyi F. Jimoh, Tracey J. Brown, Diane Bunn, Lee Hooper

**Affiliations:** 1Norwich Medical School, University of East Anglia, Norwich Research Park, Norwich NR4 7TJ, UK; tracey.j.brown@uea.ac.uk (T.J.B.); l.hooper@uea.ac.uk (L.H.); 2School of Health Sciences, University of East Anglia, Norwich Research Park, Norwich NR4 7TJ, UK; d.bunn@uea.ac.uk

**Keywords:** drinking, drinking behavior, beverages, dehydration, aged, long-term care facilities, residential care, dementia, osmolar concentration

## Abstract

Low-intake dehydration, due to insufficient beverage intake, is common in older people and associated with increased mortality and morbidity. We aimed to document the drinking patterns of older adults living in long-term care and compared patterns in those drinking well with those not drinking enough. One-hundred-and-eighty-eight people aged ≥ 65 years living in 56 UK long-term care homes were interviewed and hydration status was assessed in the Dehydration Recognition In our Elders (DRIE) study. In 22 DRIE residents, the Fluid Intake Study in our Elders (FISE) directly observed, weighed and recorded all drinks intake over 24 h. Twenty percent of DRIE participants and 18% of FISE participants had low-intake dehydration (serum osmolality > 300 mOsm/kg). Mean total drinks intake was 1787 mL/day (SD 693) in FISE participants (2033 ± 842 mL/day in men; 1748 ± 684 mL/day in women). Most drinks intake was between meals (59%, including 10% with medications). Twelve (55%) FISE participants achieved European Food Safety Authority drinks goals (3/6 men drank ≥ 2.0 L/day, 9/16 women drank ≥ 1.6 L/day). Those drinking well were offered beverages more frequently and drank more with medications and before breakfast (beverage variety did not differ). Promising strategies to support healthy drinking include offering drinks more frequently, particularly before and during breakfast and with medication.

## 1. Introduction

Despite decades of good intentions and policy to reduce dehydration in older adults, at least one in five older adults living in long-term care in the US and UK has low-intake dehydration [1,2,3,4]. Low-intake dehydration is due to not drinking enough to replace obligatory fluid losses and is characterized by raised serum osmolality. While we do not yet have high quality randomized controlled trials assessing the health impacts of reversing or preventing low-intake dehydration [5,6,7], high quality (and well adjusted) observational studies consistently suggest that low fluid intakes (poor drinking), leading to raised serum osmolality and tonicity, are associated with higher mortality and greater risk of disability in older adults [8,9,10,11,12]. Serum osmolality >300 mOsm/kg in older adults equates to low-intake dehydration (also called water-loss or intracellular dehydration) [13,14].

When we drink too little fluid, our extracellular fluid volume falls and its concentration (osmolality) rises [15]. To equilibrate osmolality, fluid moves into the serum from within cells, limiting volume reduction of the extracellular fluid, but causing cell shrinkage and raising intracellular osmolality. Cell membrane osmoreceptors trigger thirst and vasopressin release to encourage drinking and limit renal fluid losses. The basic physiological mechanisms of thirst and renal concentration ensure that fluid intake and output are balanced, but age-related changes in these mechanisms increase the risk of low-intake dehydration, making low-intake dehydration more likely as we age [3,16,17]. Older adults also often choose to reduce their fluid intakes to help control continence, minimize trips to the toilet, and reduce carer burden [17]. Recent studies in the UK and US have independently suggested that dehydration is more common in older adults living in long-term care who have cognitive impairment, renal impairment or diabetes [2,3,4]. Low-intake dehydration is distinct from excessive fluid loss through vomiting or diarrhea, when serum osmolality does not rise (as both fluid and electrolytes are lost)—the main problem is low extracellular fluid volume (fluid depletion or hypovolemia).

Recommendations for total water intake (fluid from food and drinks) vary within and between countries, ranging from 1.0 to 3.7 L/day [15,18,19,20]. US Institute of Medicine (IOM) total water adequate intake (AI) recommendations are 3.7 L/day for men and 2.7 L/day for women for adults of all ages [15], but Long-Term Care Facility Resident Assessment Instrument (RAI) recommendations are 1.5 L/day [20]. The European Food Safety Authority (EFSA) recommends a total water intake of 2.5 L/day for men and 2.0 L/day for women [19]. EFSA did not make a formal recommendation on drinks intakes, but suggests that foods contribute ~20% of total water intake, implying men should drink 2 L/day and women 1.6 L/day [19,21]. This wide variety in recommendations for older adults reflects limited underpinning research on population-level or individualized fluid needs of older adults [17].

Understanding characteristics and patterns of drinking and beverages associated with better fluid intakes and reducing low-intake dehydration prevalence in long-term care may help us to identify ways to support older adults to drink well and remain hydrated. We aimed to document how much, what and when older people living in long-term care were drinking, using direct observation and interviews. We compared patterns of drinking in participants drinking enough, with those drinking too little (less than the EFSA recommendations of 1.6 L drinks/day in women, 2.0 L/day in men) to suggest interventions that may help to improve drinking in older adults.

## 2. Materials and Methods

Dehydration Recognition In our Elders (DRIE) methods have been fully described [3,22,23] and are summarized here. The Fluid Intake Study in our Elders (FISE) was a sub-study of DRIE conducted exclusively with DRIE participants. Development and assessment of the FISE Drinks Diary (a tool for drinks self-reporting by older adults in long-term care) has been previously described [24,25]. DRIE data were collected from April 2012 to August 2013 and FISE data from May to July 2013. DRIE was approved by the UK National Research Ethics Service Committee London–East Research Ethics committee (11/LO/1997, December 2011) and FISE by the Faculty of Medicine and Health Sciences Research Ethics Committee, University of East Anglia (2012/2013-47, April 2013). Procedures for both studies were in accordance with the ethical standards of the Helsinki Declaration. DRIE study registration: Research Register for Social Care, www.researchregister.org.uk, January 2012: 122273.

### 2.1. DRIE

DRIE included people aged ≥ 65 years with written informed consent or consultee agreement, living in long-term care (residential, nursing and specialist dementia homes) in Norfolk or Suffolk, UK. We excluded those receiving palliative care or with renal or cardiac failure. The study was powered for the assessment of diagnostic accuracy of signs of hydration status [22,26]. We aimed for a representative sample of residents, while recognizing we were likely to include a higher proportion of the most physically and cognitively able. Resident interviews included the Mini-Mental State Examination (MMSE [27]), physical assessments for potential signs and symptoms of dehydration such as skin turgor and skin dryness, and a venous blood sample. We observed one drink taken by each resident where we could, in the day-room, the resident’s own room or during a meal. Non-fasting venous 3.5 mL blood samples were collected using needle and syringe, transferred to BD vacutainers® (Becton, Dickinson and Company, Franklin Lakes, NJ, USA) serum separation tubes (SST), inverted several times, stored in a temperature-controlled box, and delivered to the Department of Laboratory Medicine, Norfolk and Norwich University Hospitals (NNUH) Trust (Norfolk, UK) within 4 h of collection. Samples were used to assess serum osmolality (by freezing point depression; model 2020; Advance Instruments, repeatability ± 3 (SD1) mmol/kg in the 0–400 mmol region, laboratory CV 0.9%) [22,23]), as well as urea and electrolytes, glucose, and creatinine and analyzed on arrival. Hydration status was based on serum osmolality: normally hydrated (serum osmolality from 275 to <295 mOsm/kg), impending dehydration (295–300 mOsm/kg), or current dehydration (>300 mOsm/kg) [13,14]. Following the interview, care staff provided information on recent, current and chronic illnesses, health care contacts, medications, weight history, functional status (Barthel Index), risk factors for poor food and fluid intake or increased fluid requirements. We also elicited the usual drinks schedule for each care home. Specific questions asked are stated with responses in the results.

### 2.2. FISE

In 14 care homes, DRIE participants who had provided their own informed consent, had an MMSE score ≥ 20 and were able to write were asked whether they would like to take part in FISE. Further written informed consent was obtained from all FISE participants, including permission to access DRIE data. During one 24-h period, a researcher (OFJ) directly and continuously observed beverage (drinks) intake from 6 a.m. to 10 p.m. Both participants and night staff were asked about beverages taken from 10 p.m. to 6 a.m. All drinks served at meal times, during formal drinks rounds, with medications, and at other non-meal times were recorded, as were drinks that residents helped themselves to (from their own supplies, water coolers and faucets, drinks machines, and tea/coffee-making facilities where available). Drinks included water, flavored water, milk, flavored milk, fruit juice, squash (a fruit-flavoured concentrated syrup used to flavour water, commonly containing sugar or sweetener), Oxo and Bovril, tea, coffee, drinking chocolate, wine, beer, spirits, liquid supplements, and drinks given with medications. All drinks within the observation period were weighed in the vessel in which they were provided before and after the drinking episode, using a calibrated, digital scale (WeiHeng Electronic Scale: WH-B05). Drinks intake was recorded in mL/24 h (we assumed 1g = 1 mL for all drinks). Where residents had a personal jug, bottle or can of beverage, these were weighed before and after the observation period or when supplies were refreshed. During this 24-h period, participants completed their own Drinks Diary [24,25]. Study size was limited by researcher time available for this intensive unfunded study.

### 2.3. Data Analysis

DRIE data were analyzed using Microsoft Excel, FISE data using Excel and IBM SPSS Statistics, version 25 (IBM, Portsmouth, Hants, UK). Descriptive statistics summarized the demographic information and fluid intake. For FISE, normality was evaluated for each variable using the Shapiro–Wilk test, which is appropriate for small samples [28]. The two-sample *t*-test was used to compare fluid intake between genders and the Pearson correlation test to assess relationships between fluid intake and other variables (types of drinks, number of drinks and cognition). We did not adjust for potential confounders. Participants with intakes below EFSA recommendations (1.6 L/day of drinks for women and 2 L/day for men) were noted [19]. We chose EFSA drinks intake recommendations as our standard as they fell between the IOM and RAI recommendations [15,19,20] and split the FISE population evenly, providing the greatest power in comparing the two groups. Statistical significance was set at *p* < 0.05.

## 3. Results

### 3.1. Study Flow and Participant Characteristics

DRIE baseline interviews took place in 56 long-term care homes housing 1816 residents (see Figure 1 for study flow). We initiated research interviews with 232 individuals and obtained venous blood samples for 201. Of these, 188 participants had data that could be included in this paper. Some residents did not complete the interview or declined specific questions, so actual numbers replying are noted in the results for each question. FISE took place in 14 of the 56 DRIE care homes. Of the 62 DRIE participants residing in these homes, 32 were eligible for FISE as their MMSE score was ≥ 20 and they were able to write (see Figure 1). Ten of the 32 residents approached declined and 22 consented and took part in FISE (response rate 68.8%, reasons given in Figure 1).

The 188 DRIE participants were aged 65 to 105 (mean 85.7, SD7.8) years, 66% were female, mean BMI 25.8 kg/m^2^, see Table 1. Cognitive status in the 180 participants for whom we could assess MMSE varied from cognitively normal (MMSE 24–30) to severe impairment (MMSE < 10); 54% had cognitive impairment (MMSE < 24). Mean MMSE was 21.8, SD5.7, but was not entirely normally distributed (median 23, range 0–30, IQR 18 to 26). Functional status varied from severely limited to functionally normal (mean Barthel Index 67.4, SD26.1, median 75, range 0–100, IQR 50–90). FISE participants were cognitively and functionally more able than DRIE participants (mean MMSE 25.0, SD4.4; mean Barthel Index 79.1, SD23.6), as expected from the FISE inclusion criteria. FISE and DRIE participants were similar in age, BMI and serum osmolality (Table 1).

### 3.2. Drinks Intake and Patterns

Details of quantity, types, numbers, timing and variety of drinks consumed are reported in the Appendix A, and the key results are summarized here.

#### 3.2.1. Hydration and Drinks Intake

Twenty percent of DRIE participants were dehydrated (serum osmolality > 300 mOsm/kg) and 28% had impending dehydration (295 to 300 mOsm/kg), while 52% were well hydrated (serum osmolality 275 to < 295mOsm/kg) [3]. FISE participants were similar (18% dehydrated and 59% well hydrated) during their DRIE interviews, but hydration status was not reassessed at the FISE interview, usually several months later. FISE participants’ mean total drinks intake was 1787ml/day (SD693, range 810–3403) and did not differ by sex (2034 mL/day SD843 for men and 1748 mL/day SD684 for women, *p* = 0.66). Twelve of 22 (55%) FISE participants achieved the EFSA drinks goals (3/6 of men drank ≥ 2.0 L/day and 9/16 of women drank ≥ 1.6 L/day).

#### 3.2.2. Types of Drinks Enjoyed by Residents and Provided by Long-Term Care Facilities

Tea and coffee were the most popular drinks, while fruit juice, water, squash, or an alcoholic drink were each expressed as favorite drinks by > 10% of DRIE participants (Figure 2). These preferences were reflected in drinks provided by care homes, with ~90% of residents being provided with a cup of tea at some point during each day according to both DRIE (resident-reported, 90%) and FISE (directly observed, 91%) data. FISE direct observation showed that tea, water and coffee together contributed 80% (38%, 27% and 15% respectively) of total beverage intake. Fruit juice, hot milky drinks, milk, alcohol, and supplements each accounted for ≤ 6% intake.

FISE participants who met the EFSA minimum intakes drank more water (*p* = 0.002) than those who did not; other drink types were not significantly different. Water is the drink which residents were most able to access independently.

#### 3.2.3. Proportions of Drinks Consumed

FISE self-reported Drinks Diaries recorded which and what proportion of drinks were consumed. Drinks Diaries suggested that 86% of cups of tea and 85% of coffee served were completely consumed, while 42% of glasses of water and 14% of squash were finished. All fruit juice (18/18 drinks), hot milky drinks (9/9) and alcoholic drinks (3/3) provided were completely consumed.

#### 3.2.4. Numbers and Timing of Drinks—Routines

DRIE participants self-reported drinking 8.0 cups, mugs or glasses of drink/day. However, direct observation of FISE participants, likely to be more accurate, suggested a mean of 11.4 (SD2.9) drinks/day (range 6–16). The number of drinks/day of those who met the EFSA standard (12.8 SD2.4 drinks/day, range 9–16) was higher (*p* = 0.008) than those who did not (9.5 SD2.5, range 6–15). Number of drinks/day correlated positively with total drinks intake (*r* = 0.642, *p* = 0.001).

Care home staff were asked about the number and timing of drink offers to DRIE participants (Figure 3). Numbers of drinks offered peaked at mealtimes and were lowest at night; overall, 8.8 drinks/day were offered. As not all drinks offered were consumed and drinking 8.8 drinks/day was likely to be insufficient (if provided in 150 mL cups total intake would be 1.3 L/day), residents often had to help themselves to additional drinks to drink enough.

Whilst formal drink offers tended to be at mealtimes, as reflected by residents’ self-reported drinking times (Figure 4), FISE found that beverage intakes were greater at non-meal times. Direct observation showed that 59% of drinks volume was taken between meals (mean 1047 mL/day, SD433) with significantly less, 41% (mean 740 mL/day, SD298, *p* = 0.009), taken at meal times. This pattern was consistent for those who met the EFSA drinks intake standards and those who did not, irrespective of gender.

Drinks taken with medication (part of non-meal drinks intake) accounted for 10% of total drinks intake; ranging from 7 to 676 mL/24-h (Figure 5). More than 10% of drinks volume was consumed at each meal and mid-morning, mid-afternoon and evening drinks rounds. Intake over-night (4%) and between waking and breakfast (5%, when residents tell us they are most keen to drink) were minimal (Figure 5).

FISE participants’ drinks intake over 24 h of observation is presented in Figure 6, by whether participants met EFSA drinks intake recommendations or not. Drinks intake peaked at meals and drinks trolley rounds, suggesting that residents were largely dependent on care staff for drinks. FISE participants who met EFSA drinks intake standards had more to drink on average (although not statistically significantly more) at almost every drinking occasion through the day, especially overnight (*p* = 0.14), before breakfast (*p* = 0.06) and with medications (*p* = 0.05, Figure 6). To obtain this additional fluid, they are likely to have had to help themselves to some drinks.

#### 3.2.5. Variety of Drinks

Number of types of drinks/day is a measure of variety. A person provided with two cups of tea, two cups of coffee, an Ovaltine, and an orange juice was counted as having six drinks, but four types of drinks (tea, coffee, Ovaltine, orange juice). Mean intake was 3.5 drink types/day (SD1.3, range 1–7) by self-report in DRIE, while close observation in FISE suggested 4.0 drink types/day (SD1.3, range 2–7). Variety did not differ significantly between those who met the EFSA guidelines (mean 4.0 drink types/day, SD1.2) and those who did not (mean 3.8, SD1.3) in FISE (*r* = 0.4, *p* = 0.10).

Drinks observed during DRIE were largely presented in cups (45%, usual volume 150 mL), with 18% in mugs (approximately 225 mL), 22% in glasses (volumes vary), 10% in specialist drinks containers (such as spouted mugs), and 5% in teapots.

#### 3.2.6. Thirst and Knowledge of Hydration Status

Just before their blood test, DRIE participants were asked “are you currently feeling thirsty?”. There was no relationship between expressed thirst and hydration status as assessed by serum osmolality (*p* = 0.998) [3]. Thirst is not a good guide for the need to drink in older adults. Similarly, there was no relationship between staff-reported risk of dehydration or needing help with drinking and actual dehydration [3].

#### 3.2.7. Drinks Outside Routine Provision

While 75% of DRIE care homes reported that residents could help themselves to drinks if wanted and all homes stated residents could ask staff for drinks when wanted, FISE observations suggested that residents rarely asked for, or helped themselves to, drinks.

#### 3.2.8. Reasons for Cutting Down on Drinks

A third (32%, *n* = 64) of DRIE participants responded to the question “Do you always drink as much as you would like to?” with “no”. Some residents gave specific reasons for this including: concerns around getting to the toilet (*n* = 14); lack of available drinks (*n* = 6); not being interested in drinking (*n* = 5); not liking drinks provided (*n* = 4); functional drinking problems (*n* = 3); forgetting (*n* = 3); and being reluctant to ask staff for drinks (*n* = 1). When DRIE participants were (later) asked “Do you ever drink less so you won’t need to get up for the toilet in the night?”, 17% (*n* = 34) reported they sometimes drank less for this reason. When asked “Do you ever worry that you won’t be able to get to the toilet to pass urine in time?”, a quarter of DRIE participants (*n* = 51) reported they did worry and 8% (*n* = 15) reported drinking less because of this. These data confirm that worries about incontinence and toilet-related incidents can lead to reduced drinking in older adults [17] and reflect that 86 (46%) of 188 DRIE participants used some type of continence pad for urine, 92 (49%) used either a pad or a catheter and 36 were occasionally or regularly incontinent of bowels [3].

## 4. Discussion

Twelve (55%) of 22 FISE participants achieved EFSA drinks goals (3/6 men drank ≥ 2.0 L/day and 9/16 women ≥ 1.6 L/day). FISE participants were representative of the 188 DRIE participants (18% and 20% dehydrated respectively) and had a mean drinks intake of 1787 mL/day (SD693, range 810–3403). Tea and coffee were the most popular beverages in DRIE; and tea, water and coffee together contributed 80% of total beverage intake in FISE. Most drinks intake was between meals (59%, including 10% with medications), despite most drinks being offered at meals. FISE participants who met EFSA drinks goals (2 L/day in men, 1.6 L/day in women) [19] drank significantly more cold water, were offered more frequent drinks and drank more with medications and between waking and breakfast, but beverage variety did not differ.

DRIE data from 188 residents across 56 care homes, complemented and strengthened FISE’s more detailed observations in 22 DRIE older adults. Findings that 45% of FISE participants did not reach EFSA drinks goals correlated well with 48% of DRIE participants and 41% of FISE participants having impending or current low-intake dehydration. While direct observation and weighing of all drinks offered and consumed over 24 h provided a high-quality single assessment [29], several days observation to allow for day-to-day variation would have strengthened our study, and we suggest this is a research need. Equivalent studies assess drinks at meals and drinks rounds, but not other drinks or fluid given with medications, so they provide a less complete snapshot [4], or rely on carer diaries or recall which can be highly unreliable [24,30,31]. FISE residents who drank enough drank more with medications, drank more overnight and in the early morning than residents drinking less well. These were times when drinking was not monitored in other studies [4]. Drinks recall is problematic in older adults and is demonstrated by the difference between DRIE recall (8.0 drinks/day) and FISE observational data (11.4 drinks/day). Measuring serum osmolality confirmed whether residents were drinking sufficient fluid to avoid dehydration, and these observations of what, when and how much drinking occurred provide crucial information about how we may help older adults living in care homes to drink better.

We included alcohol intake within our fluid intake data—though < 10% of FISE participants had an alcoholic drink. While there is clear evidence that beer and lager are hydrating, we do not have data on spirits or wine [32]. Older adults may have medical or pharmaceutical reasons to be cautious of alcohol, but for others, beer and lager may provide additional enjoyable and hydrating beverages.

This is the first detailed description of drinks intake and patterns amongst UK long-term care residents, but fluid intake is better studied. Leiper et al. assessed water turnover in 15 long-term UK care residents using deuterium oxide as a tracer, finding a median fluid turnover of 1.6 L/day (range 1.0–2.8) in winter, which is lower than the FISE mean intake of 1787 mL of drinks/day (SD693, range 810–3403) [16]. It is difficult to compare drinks intake between studies as different assessment methodologies and intake standards are used, but there is agreement across all methods and standards that many older adults drink too little fluid [16,33,34,35,36,37]. Intake standards vary, probably as they are based on limited data. The US IOM AI is based on median intakes of healthy people assessed in national surveys while the EFSA AI is based on European population surveys of fluid intakes in people with desirable urine osmolality, and the basis of the current RAI recommendations is unclear [15,19,20]. IOM and EFSA recommendations are for all adults, regardless of age or bodyweight, which may (or may not) be appropriate for older adults. In older adults, fluid needs may also vary depending on a variety of health conditions including diabetes and renal failure [3]. Survey in Europe on Nutrition and the Elderly, a Concerted Action (SENECA), used dietary recall to assess the total daily water intake of free-living European elders (75–86 years), finding that most participants’ total water intake was below the SENECA standard of at least 1700ml/day but with large differences between countries [38]. The US National Health and Nutrition Examination Survey (NHANES) 2005–2010 24-h recall data suggested that 94.7% of older men and 82.6% of older women failed to meet the IOM fluid AI [39], while the Italian national food consumption survey, INRAN-SCAI 2005-06, reported that over 75% of adults (including older adults) did not comply with the EFSA AI [40]. Marra weighed the fluid intake of 132 US long-term care residents at meal and snack times (but not drinks between these formal drinking periods) across two 24-h periods, reporting a mean total water intake of 1147, SD433 mL/day. Almost all participants had inadequate total water intake [4]. We recently reviewed studies assessing hydration status by serum or plasma osmolality in older adults and showed high levels of low-intake dehydration across long-term care, the community and hospital settings [2]. FISE mean total drinks intake of 1787 mL/day, equivalent to 2.2 L/day total fluid intake, with 45% not meeting EFSA AIs (our findings), fits well with the body of literature.

Only 41% of drinks intake was during meals, 49% between meals and 10% with medications. In younger US populations, ~75% of fluid intake was peri-prandial [41], suggesting US drinking patterns, or those of younger generations, are distinct. US trials have introduced between-meal drinks as interventions (standard care was drinks provision at meals and on request [42,43]). It is unclear to us whether drinks-rounds between meals are currently standard practice in US long-term care, but they were consistently the norm in our 56 included UK homes.

Tea and coffee provided 53% of the total drinks intake in FISE, reflecting their popularity. Tea and coffee are commonly consumed in the UK, providing a greater proportion of drinks as we age [44,45]. The pattern of drinks intake observed in FISE, with little fluid intake at night, and greater daytime intake, falling after 6.00 p.m., was similar to that reported in institutionalized US and Taiwanese elders [33,35,46]. In younger UK adults, peak drinks intake was at 8 a.m. (mainly hot drinks and milk) and 9 p.m. (mainly alcohol) [44]. Very low alcohol intakes in UK residential care may be partly responsible for the low evening drinks intakes, but some DRIE participants reported avoiding drinks from late afternoon to prevent toilet trips at night, a worry also reported in US older adults [35].

## 5. Conclusions

Drinks intake is commonly too low in UK care homes. Comparing drinking patterns in older adults who drink well with those who drink too little, FISE and DRIE suggest practical strategies that long-term care facilities may use to increase fluid intake. Potentially useful strategies include offering drinks more frequently, particularly before and during breakfast and with medication (see Box 1). Drinks offered routinely should provide enough fluid to meet minimum requirements (this is not currently the case in many UK homes) and not rely on residents helping themselves to drinks. Future research needs to investigate the effectiveness of these strategies in improving drinks intake, hydration status (by serum osmolality) and health outcomes in older adults living in long-term care facilities.

Box 1Ten practical tips that may support older adults living in long-term care to increase their drinks intake.
Offering drinks more often through the day is likely to increase fluid intake. Earlier in the day can be more helpful than later, as evening drinks may be resisted. Don’t rely on residents helping themselves to drinks, or requesting them—residents who don’t, will drink too little.Regular drinks provision is vital, so care home staff should know the importance of not missing drinks rounds and that all residents must be offered drinks during rounds.Drinks handed to residents need to provide enough fluid to meet minimum requirements. Where small cups are used, more frequent drinks are needed to ensure adequate fluid.Promoting more fluid with medications helps to increase fluid intake, makes swallowing pills easier and reduces the side effects from some medications [47].Improving continence support and ease of access to toilets is likely to improve drinking.Personal preference is key, so noting and adapting to residents’ preferences for types and presentation of drinks is vital.Offering hot milky drinks, fruit juice and alcohol more frequently may improve drinking and enjoyment (as these drinks are often completely consumed, and are as hydrating as water, coffee and tea).Asking all residents whether they sometimes drink less than they would like to and if so, why? Individualizing care to address these factors may support drinking.Thirst, or lack of it, is not a good guide to whether older adults are drinking enough.All residents of long-term care are at risk of dehydration, but focusing particular drinking support on those with cognitive deficits and diabetes will support hydration [3].


## Figures and Tables

**Figure 1 nutrients-11-00447-f001:**
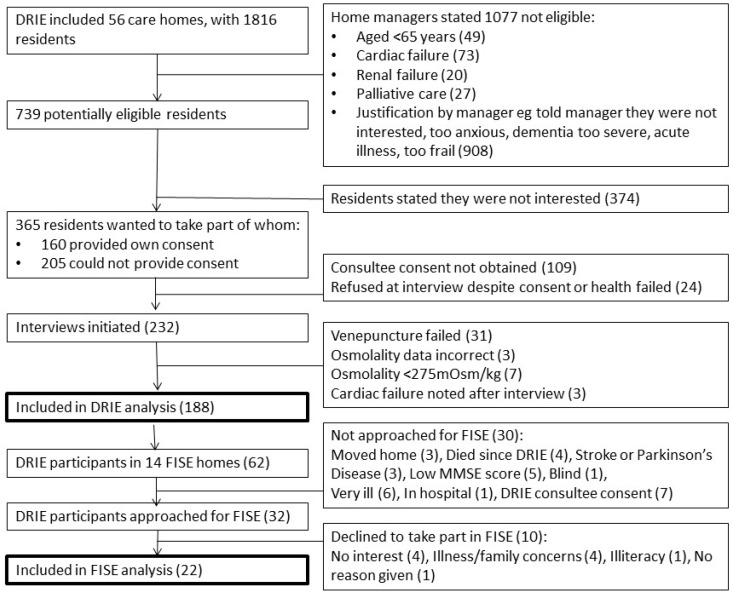
Flow of study participants (DRIE and FISE).

**Figure 2 nutrients-11-00447-f002:**
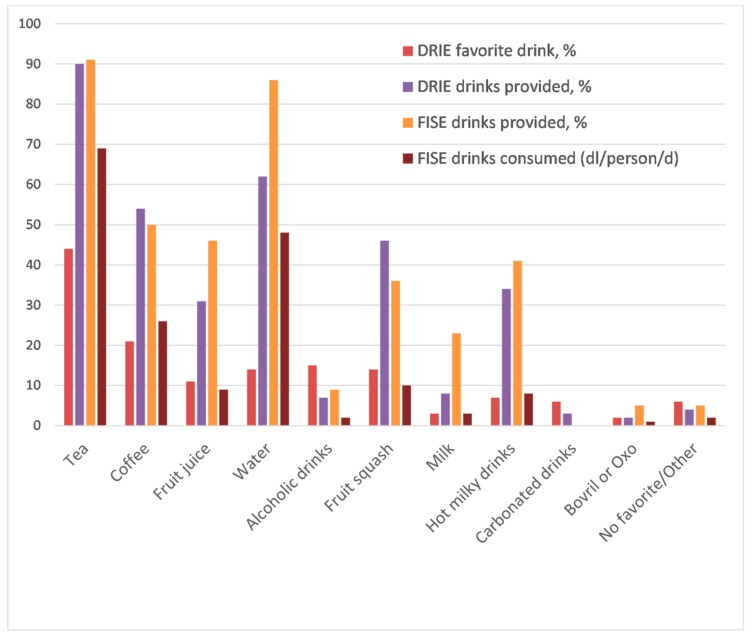
Favorite drinks and drinks provided to participants in DRIE and FISE. DRIE favorite drink: Favorite drinks as reported by 174 DRIE participants (participants could name as many favorite drinks as they liked). DRIE drinks provided: Percentage of 114 long-term care residents taking part in DRIE that self-reported receiving each drink daily. FISE drinks provided: Percentage of FISE participants receiving each type of drink over their 24-h observation period. FISE drinks consumed: Mean volume of each drink consumed by 22 FISE participants, dl/person/day.

**Figure 3 nutrients-11-00447-f003:**
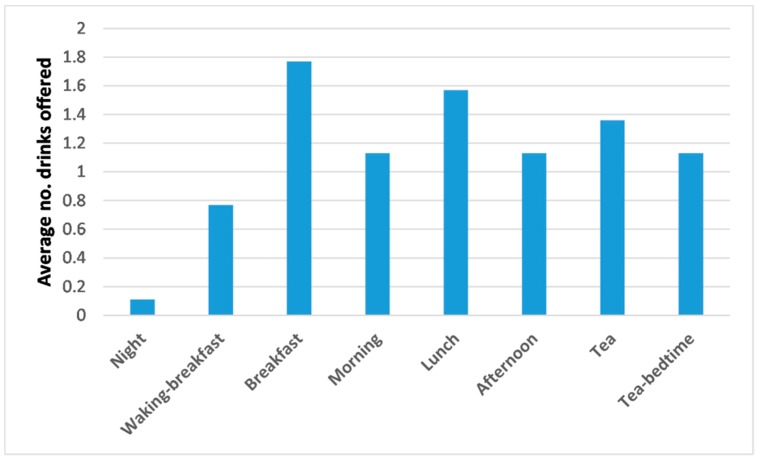
Numbers of drinks at different times of day that long-term care residents are offered formally (during a meal or a drinks trolley round), as reported by care staff during DRIE.

**Figure 4 nutrients-11-00447-f004:**
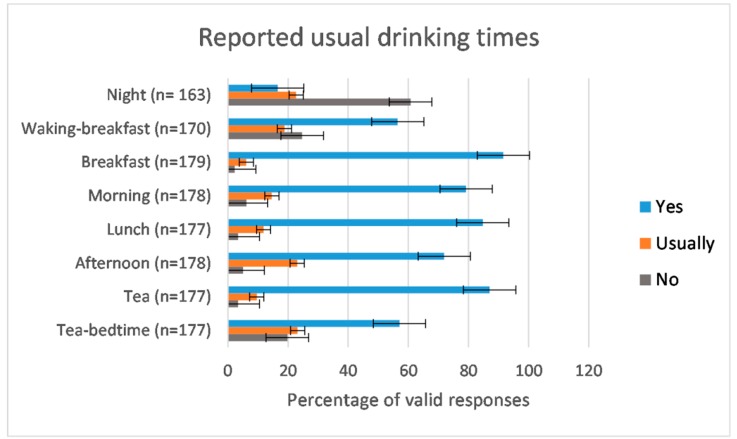
Percentage of DRIE participants responding who reported “yes, always”, “usually” or “no, never” when they were asked “Do you always have a drink during the night?” and “Do you always have a drink soon after you wake up?” etc., through the day. The number of valid responses received for each question is noted for each time of drinking. Error bars represent standard errors.

**Figure 5 nutrients-11-00447-f005:**
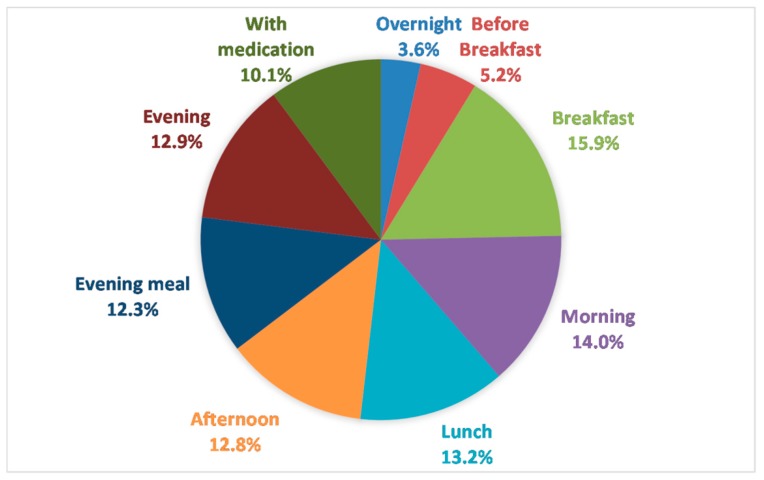
Percentage of drinks intake (volume) consumed at different time points during 24-h observation period for 22 FISE participants.

**Figure 6 nutrients-11-00447-f006:**
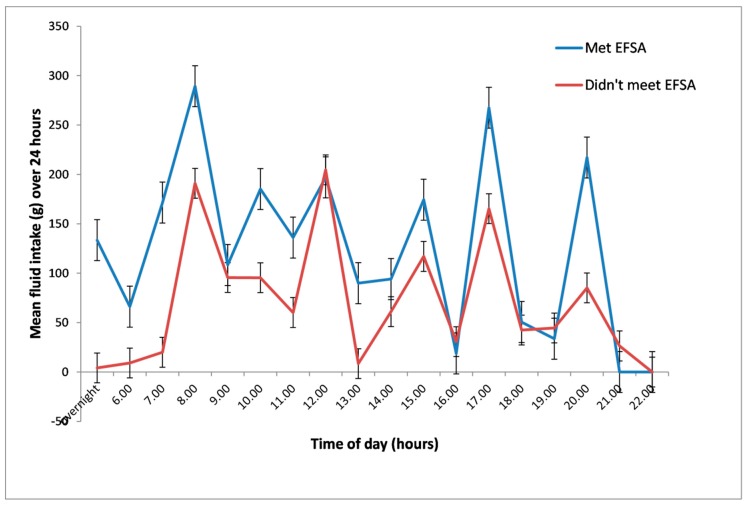
Mean drinks intake (in mL) during the day for FISE participants who met EFSA recommendations (*n* = 12) and those who did not (*n* = 10). Error bars represent standard errors.

**Table 1 nutrients-11-00447-t001:** Characteristics of DRIE and FISE participants.

Characteristic, *n* (%) Unless Otherwise Labelled	DRIE Participants *n* = 188	FISE Participants		
		All FISE Participants, *n* = 22	Low Drinks Intake (<EFSA std), *n* = 10	Good Drinks Intake (≥EFSA std), *n* = 12
**Age (years), mean ±SD (range)**	**85.7 ± 7.8** (65 to 105)	**86.8 ± 8.5** (68 to 100)	**81.9 ± 9.1** (68 to 93)	**90.8 ± 5.7** (82 to 100)
**Female**	**124 (66%)**	**16 (73%)**	**7 (70%)**	**9 (75%)**
**BMI (kg/m^2^)**, mean ± SD	**25.8 ± 5.6**	**24.7 ± 4.4**	**25.1 ± 5.5**	**24.4 ± 3.5**
Underweight, <20	32 (17%)	4 (18%)	2 (20%)	2 (17%)
Normal, 20–24.9	57 (30%)	8 (36%)	4 (40%)	4 (33%)
Overweight, 25–29.9	63 (34%)	7 (32%)	1 (10%)	6 (50%)
Obese, ≥30	36 (19%)	3 (14%)	3 (30%)	0 (0%)
**MMSE score**, mean ± SD	**21.8 ± 5.7 (*n* = 180)**	**25.0 ± 4.4**	**22.2 ± 5.0**	**27.0 ± 2.3**
Normal cognitive function (≥24)	83 (46%)	16 (73%)	5 (50%)	11 (92%)
Cognitive impairment (<24)	97 (54%)	6 (27%)	5 (50%)	1 (8%)
**Barthel Index (BI) score**, mean ± SD	**67.4 (26.1)**	**79.1 ± 23.6**	**71.0 ± 30.1**	**85.8 ± 14.7**
BI score ≥67	109 (58%)	18 (82%)	7 (70%)	11 (92%)
BI score <67	79 (42%)	4 (18%)	3 (30%)	1 (8%)
**Serum osmolality * (mOsm/kg), mean ± SD**	**293.4 ± 8.1**	**292.4 ± 9.6**	**292.5 ± 11.4**	**292.3 ± 8.4**
Dehydrated (>300 mOsm/kg)	38 (20%)	4 (18%)	2 (20%)	2 (17%)
Impending or current dehydration (≥295 mOsm/kg)	90 (48%)	9 (41%)	4 (40%)	5 (42%)

***** Serum osmolality was measured at the DRIE interview, which was often several months before the FISE interview and observations.

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
