# Peer review of "Beverage Intake and Drinking Patterns—Clues to Support Older People Living in Long-Term Care to Drink Well: DRIE and FISE Studies"

_nutrients, 2019, doi:10.3390/nu11020447_

Round 1
Reviewer 1 Report
Beverage intake and drinking patterns—clues to support older people living in long-term care to drink well: DRIE and FISE studies
Review
Overall:
The authors should be commended for undertaking an extensive piece of research and data collection. The topic is of interest and certainly has impact on the way guidelines are written for those living in long-term care facilities. Parts of the manuscript lack detail particularly in the methods section for example blood collection. There also appears to be a lack of flow in the discussion with limited reflection on the actual results. The discussion focuses on the limitations of the measures the authors used and the limited accuracy it should focus on the key outcomes of the paper as per box 10. There is a message in the data however it is not easily conveyed to the reader in its current form.
Abstract:
Line 13: Suggest rewording to clearly say the aim of the study was to document drinking patterns….
Line 14: How is drinking well determined? Euhydration and diagnostic criteria for euhydration is conflicting in many different research papers. It is important you say how drinking well is determined.
Line 15: DRIE interviewed…. A research group or area doesn’t complete the interviews. You can write 188 older people were interviewed using the DRIE tool but not the tool interviewed.
Line 22: The total volume intake that would be considered the goal intake was set by whom/how was it determined.
Introduction:
Paragraph line 54. Can you highlight some of the limitations and issues with the recommended volume of fluid intake by the respective bodies.
Is there really an optimal volume everyone should be drinking. Shouldn’t we be individualising fluid strategies based on medical history, activity levels, environment and medications.
Methods:
Line 80: please include age and criteria in abstract
Line 83: See the work by Samuel Cheuvront on diagnostic accuracy. As reference 26 isn’t published you should be using validity and published criteria
Line 92: as per above see new diagnostic criteria to set 95% CI
Were patients with cognitive and neurological conditions recruited. Patients with conditions such as parkinsons and MS may have distorted fluid intake patterns.
Clarify if venous samples were taken and how many mL’s were obtained. At what G were samples spun? Were they frozen and thawed prior to analysis? What tubes did you use to collect the blood?
Line121: why was data not adjusted for confounders – was it due to sample size?
When was the blood test taken? If it was a spot test this has a number methodological flaws.
What volume was aliquoted out for the FDP analysis. See recent paper by Robert Kenefick.
Results:
Flow chart is succinct and informative – well done. Could add some detail around why 30 of the 62 participants were not approach for FISE.
BMI wasn’t listed as an outcome measure in the methods or how it was obtained
What is meant by “not entirely normally distributed”. Data is either normally or not normally distributed.
Line 155: remove word significantly and replace with p value. Change throughout. Let the data speak for itself and leave the reader to interpret the significance
How many participants were on thickened or modified fluids?
Can you use another graphing software other than excel. The quality of the figures could be improved.
If a person is drinking 11.4 cups (assuming a cup is 250mL) than this equates to 2.8 L which would mean the vast majority would be over hydrated. What was the cup size provided?
Don’t forget fluid from food intake. If the participants are consuming water dense foods such as soups or broths then they are getting fluid via other means.
Are there any significant differences between the responses in figure 4.
3.2.6 correct thirst isn’t but perhaps the better question and data is when was the last beverage and how many mL it was and content of said beverage
3.2.7 can you run some qualitative analysis on this data. You need to provide some stats for the incontinence data
Box 10 – You don’t have the data to support point 9 or 10. If you are basing this box off other data please included appropriate references
Line 282: FISE participants came from the DRIE pool hence that findings are similar. I don’t think this is the key outcome of your trial in comparing the groups but rather stating overall levels of dehydration. If it was to compare than use statistical tools.
Line 286: Final sentence is really the importance outcome
Line 296: Please provide evidence/reference to support that 24h provided a high quality single assessment. Maybe you can reword to suggest it was the most practical method of assessment. I can’t imagine the setting would like or allow researchers to observe for multiple days plus the financial burden it would have on the research group.
The intro to the discussion and the first two paragraphs really talk down your results and basically indicate that all the measures you used are inaccurate and not the best (makes the reader question why you did the study then if non of it works). Suggest highlighting your results and make them shine and then in a limitations paragraph discuss the barriers.
Line 314 – Line 338: This paragraph reads more of a literature review rather than a discussion of your results and what your results mean. The paragraph started well and I thought this was better suited based on the above comments however it then reviewed each paper on the topic which isn’t a discussion.
Line 339: Indicate if this was in your study
Supplementary file
Include p values in 3.2 for sex difference in fluid intake
Please revise first sentence of 3.3 it doesn’t make sense
Where there reasons that the participants missed drinks?
‘Over 10% of drink volume was consumed at each meal’. Does this conflict figure 4 where the majority of participants said they consumed fluid at meal times
Figure 6 perhaps put some key time points on there e.g. lunch time or breakfast time
Of the 27% that answered yes to being thirsty how many of those were dehydrated. A correlation analysis isn’t the best tool for osmol and thirst. Suggest looking if thirst was indicative of dehydration.
Author Response
We have uploaded our replies to reviewer 1, please see attached word document.

Reviewer 2 Report
DRIE is a well done study, but FISE have a small sample. Furthermore, write-up does not adequately reflect or discuss the literature or the findings. I have specific comments:
· 22 residents are not very representative.
· Line 73: It is a bit old data.
· Line 80: I think that there is a great limitation in the results due to the exclusion of patients without renal or cardiac failure. This can condition: the basal state of the residents, the prevalence of dehydration, the fluid intake of these and the factors associated. In other studies, the low fluid intake was much bigger (Botigué, 2018; Reed ,2005 or Simmons, 2001).
· Line 86 or 94: Which are these signs and symptoms?
· Line 101: Collecting fluid intake over 24h may not reflect the fluid intake pattern of residents, since here are various organizational factors that could influence the final result or insufficient staff ratios, which tend to be more frequent on weekends.
· Lines 122-123: EFSA recommendations are not take into account the physiology, and the chronic disease of old people, often went with water restriction. EFSA also states that adequate intakes of water for older adults, therefore, should not be based solely on observed intakes, but should take into account the decreases in renal concentrating capacity with age and the decrease in thirst sensitivity. So, discuss.
· Lines 304-307: I do not know if I correctly understood this paragraph. So, did you take blood tests at a different time from the collection of liquid intake? If so, the baseline conditions of the residents could change.
· You do not expose the limitations of the FISE study in sufficient depth.
· Can you provide more information about incontinence ratios?
Author Response
Thank you for your comments and helping us to improve our paper, we have replied to your comments in the attached document.

Round 2
Reviewer 1 Report
I thank the authors for taking on board the comments from the first review. I just have some minor comments below.
Line 63-64: Your response has a reference [1] but in the manuscript doesn’t please correct accordingly
There are grammatical errors in the methods section specifically – line 81 check spacing and full stop line 97. Please check.
Graphical software could include R or prism. Quality of the figures can be at the editor’s discretion.
Figure 4 and Figure 5. Re statistics. You already have the groupings and the responses. A chi square analysis could be performed. Don’t compare to the single time of day there is no standard time of day. You just need to compare across the categories you already have with the number of valid responses.
Author Response
Please see our answers attached.

Reviewer 2 Report
I believe these revisions have radically improved the paper's argument.
Author Response
Reviewer 2
Comments and Suggestions for Authors
I believe these revisions have radically improved the paper's argument.
Authors’ response: Thank you.